# Mechanisms of Antimicrobial Peptides from Bagasse against Human Pathogenic Bacteria

**DOI:** 10.3390/antibiotics12030448

**Published:** 2023-02-23

**Authors:** Thitiporn Ditsawanon, Narumon Phaonakrob, Sittiruk Roytrakul

**Affiliations:** 1Faculty of Science and Technology, Rajabhat Rajanagarindra University, Chachoengsao 24000, Thailand; 2Functional Proteomics Technology Laboratory, National Center for Genetic Engineering and Biotechnology, National Science and Technology Development Agency, Pathum Thani 12120, Thailand

**Keywords:** AMPs, agricultural wastes, peptide purification, human pathogenic bacteria

## Abstract

Nonedible agricultural wastes (agricultural wastes, agro-industrial wastes, and fishery wastes) were chosen as potential sources of antimicrobial peptides and evaluated for antibacterial efficiency against human pathogens. Specifically, protein hydrolysates were first obtained by hydrolysis with pepsin. Filtrated peptides smaller than 3 kDa were then purified by C18 reversed-phase chromatography, cation exchange chromatography, and off-gel fractionation. NanoLC-MS/MS was used to investigate the amino acid sequences of active peptide candidates. Five candidate peptides were finally chosen for chemical synthesis and evaluation of growth inhibition against human pathogenic bacteria. Two synthetic peptides from bagasse, NLWSNEINQDMAEF (Asn-Leu-Trp-Ser-Asn-Glu-Ile-Asn-Gln-Asp-Met-Ala-Glu-Phe) and VSNCL (Val-Ser-Asn-Cys-Leu), showed the most potent antibacterial activity against three pathogens: *Pseudomonas aeruginosa*, *Bacillus subtilis*, and *Burkholderia cepacia*. The antibacterial mechanisms of these peptides were then examined using shotgun proteomics, which revealed their effects to involve both intracellular-active and membrane-active mechanisms. Further investigation and modification of peptides are needed to increase the efficiency of these peptides against human pathogens.

## 1. Introduction

Pathogenic bacteria are common causes of infectious illnesses that account for about half of human deaths worldwide. Antibiotics and synthetic medicines have been the primary tools of treatment in such illnesses, but their long-term repeated use causes development of antibiotic resistance. This is reflected in the increase of multidrug-resistant (MDR) bacteria over the past few decades. A report in the United States revealed that approximately 2.8 million people are infected and more than 35,000 people die from antibiotic-resistant pathogens each year. Thus, antimicrobial resistance is rapidly becoming an unescapable health crisis [1]. 

Among MDR bacteria, *Pseudomonas aeruginosa* is a critical opportunistic pathogen that causes nosocomial infections which lead to high mortality, especially in immunocompromised or intensive care unit patients [2,3,4]. This pathogen can resist several types of antibiotics, such as fluoroquinolones, β-lactam antibiotics, and aminoglycosides, through a variety of antibiotic-resistant mechanisms [5]. In addition to *P. aeruginosa*, *Bacillus subtilis* is another opportunistic pathogen that is resistant to several antibiotics, such as chloramphenicol, tetracycline, erythromycin, lincomycin, penicillin, and streptomycin. The conditions caused by *Bacillus subtilis* include pneumonia, bacteremia, endocarditis, and septicemia. However, this pathogen is most found in patients with compromised immune states [6,7]. Another pathogen of interest that is resistant to antibiotics is *Burkholderia cepacia*. Indeed, infections caused by this pathogen are difficult to cure because of its considerable resistance, leading to high mortality rates in cystic fibrosis patients [8].

Thus, there is a critical need to investigate new antibacterial agents so as to overcome bacterial antibiotic resistance. Antimicrobial peptides (AMPs) are natural protectors against pathogens that play roles in innate immune systems [9]. To date, more than 5000 AMPs have been found in many organisms, both prokaryotes and eukaryotes [10]. Commonly, AMPs are categorized according to their targets as being antifungal, antibacterial, antiparasitic, or antiviral [11]. They are also further subclassified into families, such as defensins, thionins, snakins, hevein-like proteins, cyclotides, and others. Generally, a slight majority of AMPs are antifungal, at 51%, whereas antibacterial and antiviral categories account for 33% and 10% of AMPs, respectively [12]. The antibacterial activity of AMPs is mostly attributable to their amphiphilic composition and high positive charge, characteristics which help the peptides attach to and enter into bacterial pathogens by making a pore in the cell membrane, leading to membrane disruption and cell lysis [13,14]. 

For this study, agricultural wastes were chosen as AMP sources because of their abundant production in association with global population growth. Agricultural wastes are produced mainly from agricultural business endeavors involving food production, agro-industries, and fisheries, and the quantity of waste produced from agricultural manufacturing is growing by 7.5% each year [15]. The negative effects of agricultural wastes on humans, animals, and the environment are considerable. In many developing countries, agricultural wastes are burnt or randomly dumped in communal areas, which inappropriate management causes pollution of the air, soil, and water [16].

Here, the aim of the study was to purify and characterize the AMPs obtained from peptic hydrolysis of agricultural wastes. The <3 kDa peptides having antibacterial activity were purified with reversed-phase chromatography, cation exchange chromatography, and off-gel fractionation. Finally, the amino acid sequences of active peptide candidates were analyzed by LC-MS/MS, while some peptides were chemically synthesized, with their antibacterial activity against human pathogenic bacteria determined and their mechanisms of action investigated via shotgun proteomics. 

## 2. Results

### 2.1. Antibacterial Efficiency of Lower than 3 kDa Protein Hydrolysates/Peptides 

Out of the fifteen waste samples (six agricultural wastes (AW1-6), seven agro-industrial wastes (IW1-7) and two fishery wastes (FW1-2)), two hydrolysates (AW6 and IW4) presented obviously higher antibacterial efficiency, with inhibitory percentages of 50% or higher for every targeted human pathogenic bacterium (except IW4 against *Bacillus subtilis*, 23.71%), as shown in Table 1.

### 2.2. Antibacterial Activity of Purified Peptides from Each Purification Step 

After antibacterial activity screening, two protein hydrolysate samples (AW6 and IW4) were selected for further successive purification by reversed-phase chromatography, cation exchange chromatography, and off-gel fractionation. Bacterial growth inhibition was evaluated in triplicate after each purification step and the effective fractions were carried through to the next step as described in Figure 1. Samples written in black were the most effective and were selected for further purification step, while samples written in light color were not selected.

For both AW6 and IW4, the unbound fraction from reversed-phase chromatography (UBR) exhibited higher inhibitory potential than the bound fraction (BR), i.e., hydrophilic fractions gave higher activity than hydrophobic fractions. After cation exchange chromatography through a HiTrap SP Sepharose FF 1 mL (Cytiva) column, peptides with high activity were found to bind to the column and named AW6 UBR-BC. Accordingly, the AW6 UBR-BC fraction was chosen to be purified by off-gel fractionation and five bioactive fractions were selected for the analysis of peptide sequences using LC-MS/MS and Mascot software.

### 2.3. Antibacterial Analysis of Synthetic Peptides

A large number of peptides were detected and analyzed for their amino acid sequences, after which five peptides with high Mascot peptide scores were selected for the evaluation of antibacterial activity (Table 2). Peptide no.3 (NLWSNEINQDMAEF) and peptide no.4 (VSNCL) showed the most effective activity against *Bacillus subtilis* and *Burkholderia cepacia*, while also respectively ranking second and third against *P. aeruginosa.* Accordingly, these two effective peptides were selected for further study of peptide-microbe mechanisms.

### 2.4. Study of Peptide-Microbe Interaction Mechanisms

Shotgun proteomics was employed for the investigation of peptide-microbe interaction mechanisms. *Pseudomonas aeruginosa*, *Bacillus subtilis*, and *Burkholderia cepacia* were treated with synthetic peptides no.3 and no.4 for 6 h, with untreated and antibiotic-treated samples used as controls. Proteins from all treated and untreated microbe samples were isolated, reduced, alkylated, tryptic digested, and injected into a LC-MS/MS instrument. The resulting MS/MS peptide data was analyzed by DecyderMS™, exported, and further analyzed by the Mascot software using protein databases available in Uniprot (https://www.uniprot.org; accessed on 6 April 2022). All proteins identified as differentially expressed between treated and untreated samples were then visualized and commonalities evaluated by Venn diagram [17]. A Venn diagram uses circle or triangle shapes to illustrate the overlap and difference between sets of data, here groups of proteins from each treatment (control, peptides, antibiotics). Circles that overlap include the same proteins while circles that do not overlap do not share any proteins.

As demonstrated in Figure 2, the total count of differentially expressed proteins found in *P. aeruginosa* was 198 when treated with peptide no.3 and 248 with peptide no.4. Of those, 136 and 186 proteins were unique to the peptide treatment, while a small number were also perturbed upon oxycline treatment. Meanwhile, a total 99 and 98 differentially expressed proteins were identified in *Bacillus subtilis* treated with peptides no.3 and no.4, of which 39 and 43 were unique to the respective peptide treatment while ten or fewer were shared with any antibiotic treatment (ampicillin, kanamycin, or oxycline). Finally, there were 4855 and 2947 differentially expressed proteins respectively found in *Burkholderia cepacia* treated with peptide no.3 and with peptide no.4. Among those, 436 and 132 were unique to the peptide treatment, while 129 proteins were commonly perturbed by peptide no.3, kanamycin, and oxycline; 109 by peptide no.3, no.4, and oxycline; and 156 proteins by peptide no.4 and kanamycin. 

From the Venn diagram in Figure 2, proteins of treated bacterial pathogens can be organized by the overlapped areas between peptides and antibiotics (non-overlapped with control), as shown in Table 3. 

### 2.5. Functional Classification of Differentially Expressed Proteins 

The sets of differentially expressed proteins unique to peptide treatments were further analyzed for functional relevance (Gene Ontology terms) through UniProt, an approachable database of protein sequence and functional information (Figure 3).

For *P. aeruginosa* treated with peptide no.3, the category most highly represented among differentially expressed proteins was regulation of transcription and translation (28.57%), with other prominent terms being ATP binding, lipid and protein metabolism, DNA synthesis, amino acid synthesis, chemotaxis, cell membrane component, pilus assembly, and signal transduction, as shown in Figure 3a. These results suggest that the antimicrobial mechanism of peptide no.3 in *P. aeruginosa* primarily concerns intracellular and biosynthesis processes. Meanwhile, treatment with peptide no.4 impacted proteins related to the cell membrane (21.59%), regulation of transcription (19.32%), and to a lesser degree bio-compound synthesis, DNA activity, biofilm, and toxin activity (Figure 3b). These results imply that peptide no.4 interacts with proteins involved in the cell membrane and in intracellular processes.

Similarly, in *B. subtilis* treated with peptide no.3 (Figure 3c), the most-represented terms were related to the cell membrane, proteolysis, and fatty acid and DNA synthesis. When treated with peptide no.4 (Figure 3d), the largest category of differentially expressed proteins related to regulation of transcription (42.31%), followed by terms pertaining to the cell membrane, transmembrane transport, carbohydrate metabolism, amino acid synthesis, and DNA replication. These results imply that, in *B. subtilis*, both peptide no.3 and no.4 interact with proteins related to intracellular biosynthesis and transmembrane activity.

Finally, the proteins differentially expressed in *Burkholderia cepacia* after treatment with peptide no.3 primarily were related to regulation of transcription (19.86%, Figure 3e), while treatment with peptide no.4 altered most impacted proteins related to transmembrane transport (27.66%, Figure 3f). The next most abundant category for both treatments was bio-compound biosynthesis. Thus, it is possible that, in *Burkholderia cepacia*, peptides no.3 and no.4 may both cause cell leakage and affect bio-compound metabolism, i.e., metabolism of lipids, carbohydrates, proteins, and DNA. 

## 3. Discussion

### 3.1. Protein Hydrolysate Preparation and Antibacterial Activity Screening

In the initial protein extraction of this study, sample proteins were extracted with 0.05 M sodium acetate, pH 4, in compliance with previous reports that mild-acidic extraction is suitable for protein extraction from plants and can minimize oxidation, phenolic compound polymerization, and irreversible protein binding [18]. Lay et al. [19] likewise extracted protein from ornamental tobacco and petunia with 50 mm sulfuric acid. Pickardt et al. (2009) [18] found that increasing the concentration of sodium chloride enhanced relative protein yield, while Taniguchi et al. [20] adjusted the pH to 2.0 with 1M HCl before hydrolysis with pepsin. In the present study, the obtained peptides were heated in an autoclave (121 °C for 15 min) to remove heat-intolerant and heat-labile proteins, congruent with the work of Lay et al. [21].

According to the report of Broekaert et al. [22], plant antimicrobial peptides (AMPs) are normally smaller than 10 kDa. Accordingly, we used a semipermeable membrane to filtrate the protein hydrolysates before purification. Such short peptides also allow for cost saving in synthesis. Gordon et al. [23] reported that peptides of less than ten amino acids are typically a good option with respect to synthesis cost. Therefore, in the present study, the Vivaspin 20 3 kDa MWCO was used to retain only peptides of less than 3 kDa (approximately 27–28 amino acids). 

Evaluation of the inhibitory percentages of protein hydrolysate samples against human pathogenic bacteria (Table 1) revealed AW6 (protein hydrolysate from bagasse) to rank first or second against every tested pathogen. These results conform to the report of Velazquez-Martinez et al. [24] that sugarcane bagasse with 2.2% crude protein has strong antimicrobial properties against *Escherichia coli*, *Bacillus cereus*, and *Staphylococcus aureus*. Additionally, a coconut residue sample (IW4) ranked second, which also conforms to a prior report that coconut AMPs have extremely efficient antimicrobial activity against both Gram-positive and Gram-negative pathogenic bacteria (namely *E. coli*, *B. subtilis*, *S. aureus*, and *P. aeruginosa*) [25]. 

### 3.2. Protein Hydrolysate and Peptide Purification

Methods of protein purification typically rely on intrinsic physio-biochemical characteristics including net charge, size, thermostability, and solvent tolerance. Scott et al. [26] emphasized that some mixtures of proteins from waste samples should be segregated according to the presence of polar and nonpolar side groups, while for others, basic and acidic amino acids may instead be separated. In addition, complex mixtures of amino acids can also be separated using chromatographic purification approaches. Ion exchange chromatography is therefore an important method in the separation of charged peptides from complex mixtures. 

After identifying two hydrolysates as having effective antibacterial activity (AW1 and IW4) against human pathogenic bacteria, we applied reversed-phase chromatography to separate hydrophobic and hydrophilic fractions, followed by cation exchange chromatography and then off-gel fractionation (pI-based purification), with the antibacterial activity of all fractions being assayed at each stage and only active fractions carried through to the next purification. Thus, with each purification step, the scope of the bioactive peptides was further narrowed. After reversed-phase chromatography, two UBR samples (AW6 UBR and IW4 UBR) exhibited high antibacterial activity. Subsequently, after cation exchange chromatography, only AW6 UBR-BC showed antibacterial activity. Finally, after pI-based purification, five samples (wells 5, 6, 7, 13, and 18) were selected for further analysis by LC-MS/MS and the Mascot software. A final five peptides with high peptide scores were selected for chemical synthesis and an investigation of their mechanisms of action against human pathogens.

### 3.3. Peptide-Microbe Interaction Mechanisms

As illustrated in Figure 2 and Table 3, the majority of proteins differentially expressed upon peptide treatment in *P. aeruginosa* and *B. subtilis* were unique (that is, not also perturbed upon antibiotic treatment). Accordingly, it could be concluded that the tested peptides have unique mechanisms of action against these bacteria. Conversely, a greater proportion of proteins were mutually perturbed by peptide and antibiotic treatments in *B. cepacia*, suggesting that some parts of the peptides act like antibiotics in *B. cepacia*. 

Subsequent functional analysis and classification of the uniquely differentially expressed proteins using UniProt revealed the two AMPs to both have a dual mode of action. That is to say, some part of peptides no.3 and no.4 acts as an intracellular-active AMP and some part acts as a transmembrane-active AMP. Intracellular-active AMPs can kill or inhibit growth of microbial cells without creating membrane disruption through their interaction with intracellular targets, such as DNA, RNA, proteins, or other molecules [27], while membrane-active AMPs, in the proper conditions, create channels or transmembrane pores that cause cell leakage and an outflow of intracellular molecules, ultimately leading to cell death [28].

Numerous known AMPs have only one mode of action. For example, apidaecin has shown intracellular activity with nonmembrane disruption at every tested concentration and condition [29]. Nevertheless, dual mechanisms are also reported for many AMPs, often contingent on peptide concentration. Generally, concentrations higher than the minimum inhibitory concentration (MIC) result in membrane lysis with the AMP acting as a detergent, whereas concentrations lower than the MIC lead to membrane penetration and effects on intracellular macromolecules [30], ultimately resulting in the inhibition of bacterial growth and bacterial death. In addition to concentration, the mode of peptide action is also dependent on peptide structure, peptide concentration, lipid membrane composition, temperature, and pH [31]. 

Interestingly, the results of this work show peptide no.3 and no.4 to primarily act as intracellular-active AMPs, but also to have some factor that affects the cell wall and cell membrane. This effect is similar to some previous observations. Shi et al. [32] reported that melittin, an AMP from *Apis mellifera*, interrupts the membrane, making holes and causing leakage of cytoplasm, and that it may also inhibit the biosynthesis of both DNA and proteins. The bactericidal peptide indolicidin has also shown dual mechanisms of antimicrobial action. In aqueous solution, its structure is amphipathic and globular, while in a lipid environment it takes on opposing properties. These two structures then have different effects, the former binding DNA and the latter interacting with lipid bilayers [33].

## 4. Materials and Methods

### 4.1. Time and Place of Research

This research was carried out during 2021–2022 at the Functional Proteomics Technology Laboratory, National Center for Genetics Engineering and Biotechnology, National Science and Technology Development Agency, Pathumthani, Thailand.

### 4.2. Sample Collection

Six agricultural wastes, seven agro-industrial wastes, and two fishery wastes were collected from different parts of Thailand (Table 4 and Figure 4).

### 4.3. Preparation of Lower than 3 kDa Protein Hydrolysates

Total proteins from all collected samples were extracted by mechanical shaking with 0.05 M sodium acetate buffer, pH 4.0, at 25 ± 2 °C for 1 h and then heating at 121 °C for 15 min. The protein concentration of each sample was evaluated by Lowry assay [34] using bovine serum albumin (BSA) as the protein standard. Next, the obtained total proteins were hydrolyzed with pepsin (Sigma-Aldrich, St. Luis, MO, USA) at a ratio of 1:25 by shaking at 200 rpm for 12 h at 37 °C, then boiled for 15 min to terminate the reactions. The supernatants of the crude hydrolysates were collected by centrifugation at 10,000× *g* for 10 min at 25 ± 2 °C, then five-times diluted with 0.05 M sodium acetate buffer, pH 4.0. The diluted hydrolysates were finally filtrated through a semipermeable membrane (Vivaspin 20, 3 kDa MWCO, GE Healthcare, Chicago, UK), with the peptides lower than 3 kDa in size being collected and kept at −20 °C until use.

### 4.4. Human Pathogenic Bacteria and Antimicrobial Activity Assay

Three human pathogenic bacteria were selected for investigation, namely *Bacillus subtilis* ATCC6633, *Pseudomonas aeruginosa* ATCC28753, and *Burkholderia cepacia* ATCC25416. 

In evaluating antibacterial activity by the broth dilution assay, human pathogenic bacteria were first prepared in tryptic soy agar (TSA) (Difco BBL, Sparks, MD, USA) for 24 h at 28 °C. Then, a single colony was cultured in tryptic soy broth (TSB) (Difco BBL, Sparks, MD, USA) for 12–16 h to reach an inoculum of 0.05 at OD_600_. The antibacterial activity of lower than 3 kDa protein hydrolysates/peptides was then determined against the pathogens in triplicate using 96-well microplates, with bacteria in TSB, phosphate-buffered saline (PBS), and antibiotic treatment (ampicillin and kanamycin) used as controls. The final concentration of antibiotics and protein hydrolysates/peptides was 100 µg/mL. After incubation for 0, 2, 4, and 6 h, the OD_600_ was documented using a microplate reader (Synergy H1 Hybrid Multi-Mode Reader, Biotek, Winusky, VT, USA). 

### 4.5. Experimental and Statistical Design 

A completely randomized design was used: protein hydrolysates/peptides from six agricultural wastes, seven agro-industrial wastes, and two fishery wastes as treatments; and human pathogenic bacteria *P. aeruginosa* ATCC28753, *Bacillus subtilis* ATCC6633, and *Burkholderia cepacia* ATCC25416 grown in tryptic soy broth (TSB) (Difco BBL, Sparks, MD, USA) at 28 °C in 96-well plates as experimental units. Experiments were conducted in triplicate and all the results shown in this study are presented as mean ± standard deviation (SD).

### 4.6. Reversed-Phase Chromatography (1st Step of Peptide Purification)

The active peptides were first purified by reversed-phase chromatography through a Delta-Pak C18 column (100 Å, 3.9 mm × 150 mm; Interlink Scientific Services Ltd., Kent, UK), initially equilibrated with 0.1% trifluoroacetic acid (TFA) in acetonitrile (ACN). The column was washed with 0.1% TFA in sterile water, after which sample in 0.1% TFA was applied to bind the column. The hydrophilic fraction (coded UBR; unbound fraction of reversed-phase chromatography) in the column was eluted with 0.1% TFA in sterile water. Afterward, the hydrophobic fraction (coded BR; bound fraction of reversed-phase chromatography) was eluted stepwise with 0.1% TFA in ACN. The flow rate of all steps in the reversed-phase chromatography approach was 1 mL/min. Both UBR and BR fractions of each sample were used in the further determination of antimicrobial activity. 

### 4.7. Cation Exchange Chromatography (2nd Step of Peptide Purification)

The active fractions from reversed-phase chromatography were subsequently ion-separated by cation exchange chromatography through an AKTA^TM^ start system (GE Healthcare, Chicago, IL, USA). Peptide samples were adjusted to pH 4 and applied via a HiTrap SP Sepharose FF 1 mL cation chromatography exchange column (Cytiva, Marlborough, MA, USA) with flow rate of 1 mL/min and fraction volume of 1 mL. Then, the column was washed out with 50 mM NaOAc, pH 4, and the unbound fraction of cation exchange chromatography (coded UBC) was gathered. Afterward, the bound fraction of cation exchange chromatography (coded BC) was eluted by a gradient of 0–100% of 1M NaCl. Each fraction was buffer-exchanged from 50 mM NaOAc, pH 4 with a gradient of 1 M NaCl to sterile water through a P-6 desalting column (Bio-Rad Laboratories, Inc., Heracles, CA, USA) before antibacterial activity evaluation.

### 4.8. Off-Gel Fractionation (3rd Step of Peptide Purification)

The active fractions from cation exchange chromatography were separated according to their isoelectric points (pIs) using an Agilent 3100 OFFGEL Fractionator with an 18-well (18 cm) plate and pH gradient of 3–10. The electrode was placed at the point of lowest pH and then swell buffer was pipetted onto it. The 18-well plate was placed next to the electrode, and another electrode was placed at the end of the multi-well with the highest pH. Then, voltage was applied, and the separation process was conducted according to the supplier’s protocol, as given in Table 5. The active peptides after fractionation were further analyzed by nanoLC-MS/MS and the Mascot software (Matrix Science, London, UK) [35].

### 4.9. Peptide Synthesis 

Selected peptides were chemically synthesized via solid-phase peptide synthesis as reported by Hansen and Oddo [36]. First, resin was loaded into a hand coupling vessel and washed three times with N,N-dimethylformamide (DMF, anhydrous 99.8%), after which 9-fluorenyl methyl oxycarbonyl (Fmoc) in the amino group on the amino acid was deprotected by adding 5 mL of 20% piperidine. After washing another five times with DMF, the activated amino acid solubilized with coupling agent and 20% collidine in DMF was added and mixed for 15 min. The deprotecting and coupling steps were repeated until finishing the last amino acid in the sequence. Then, the Fmoc group was cleaved by treating the peptide on resin with 20% piperidine in DMF. Finally, the synthesized peptide samples were partially purified with Sep-Pak C18 Cartridges before antimicrobial activity determination.

### 4.10. Study of Peptide-Microbe Interaction Mechanisms Using Proteomics

The protein profiles of human pathogenic bacteria with and without effective peptide treatment (6 h) were determined by LC-MS/MS. After reduction with dithiothreitol, alkylation with iodoacetamide, and tryptic digestion, peptides were injected into an Ultimate 3000 Nano/Capillary LC system (Thermo Scientific, Waltham, MA, USA) coupled to a Hybrid quadrupole Q-Tof impact II™ (Bruker Daltonics) equipped with a Nano-captive spray ion source. The MS/MS spectra results were analyzed with MaxQuant 1.6.6.0 [37], and the Andromeda search engine was used to correlate MS/MS spectra to the Uniprot database. Subsequently, all data were simply visualized as a set of relationships by Venn diagram [17], and the functional relevance of uniquely differentially expressed proteins was identified using UniProt (uniprot.org/blast; accessed on 13 May 2022).

## 5. Conclusions

Protein hydrolysates from bagasse (AW6) and coconut residue (IW4) can be used to inhibit the growth of *Pseudomonas aeruginosa*, *Bacillus subtilis*, and *Burkholderia cepacia*. After three different purification steps, two peptides sourced from bagasse hydrolysate, namely NLWSNEINQDMAEF (peptide no.3) and VSNCL (peptide no.4), showed antibacterial activity against all three bacterial species. Mechanistically, both peptides may inhibit bacterial growth via interfering with the bacterial cell membrane as well as targeting intracellular biomolecules and inhibiting metabolic processes. Further study remains needed to customize AMPs by changing the amino acids in some positions for greater effectiveness and stability, and checking their cytotoxicity against human cell lines.

## Figures and Tables

**Figure 1 antibiotics-12-00448-f001:**
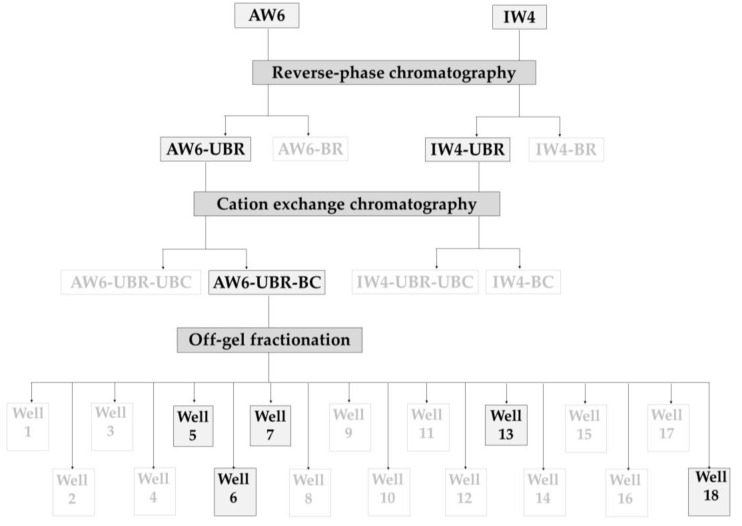
Flow chart of the effective fractions after purification steps.

**Figure 2 antibiotics-12-00448-f002:**
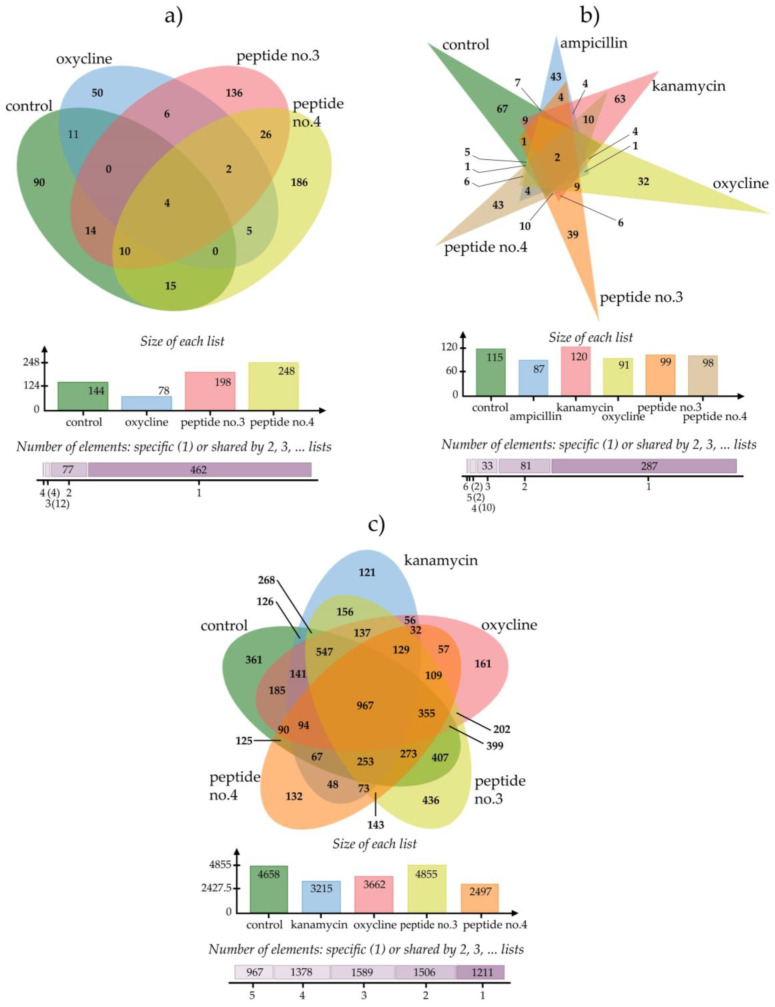
Venn diagrams of differentially expressed proteins in bacterial pathogens after treatment with peptides and antibiotics: (**a**) *P. aeruginosa*, (**b**) *Bacillus subtilis*, and (**c**) *Burkholderia cepacia*.

**Figure 3 antibiotics-12-00448-f003:**
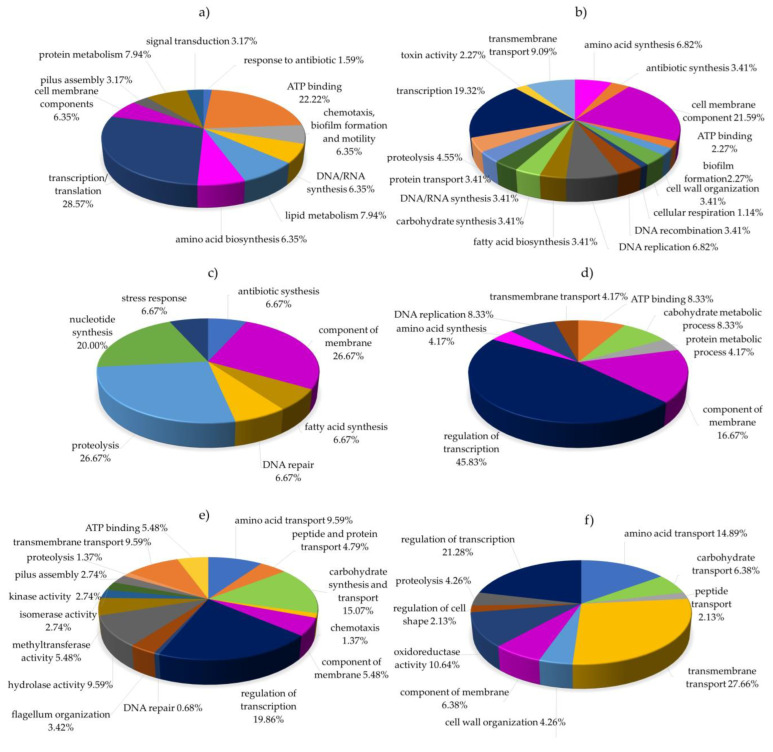
Functional classification distributions of differentially expressed proteins in peptide-treated human pathogens: (**a**) *P. aeruginosa* treated with peptide no.3, (**b**) *P. aeruginosa* treated with peptide no.4, (**c**) *Bacillus subtilis* treated with peptide no.3, (**d**) *Bacillus subtilis* treated with peptide no.4, (**e**) *Burkholderia cepacia* treated with peptide no.3, and (**f**) *Burkholderia cepacia* treated with peptide no.4.

**Figure 4 antibiotics-12-00448-f004:**
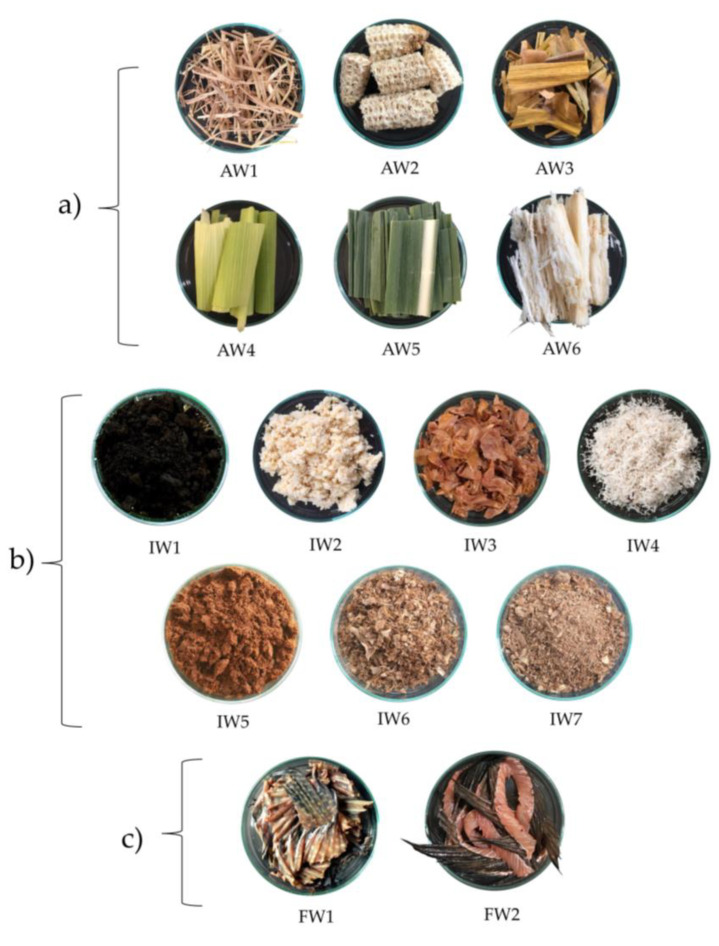
The three groups of waste samples: (**a**) agricultural wastes (AW), (**b**) agro-industrial wastes (IW), and (**c**) fishery wastes (FW).

**Table 1 antibiotics-12-00448-t001:** Ranking of inhibitory percentage of lower 3 kDa hydrolysate samples against human pathogenic bacterial growth at 6 h after treatment.

Antibacterial Activity Ranking	Inhibitory Percentage of Target Human Pathogenic Bacteria *					
	*Pseudomonas aeruginosa*		*Bacillus subtilis*		*Burkholderia cepacia*	
1	79.49 ± 0.49 ^a^	AW6	52.73 ± 4.58 ^a^	AW6	85.66 ± 0.91 ^a^	AW6
2	56.33 ± 8.79 ^b^	IW4	23.71 ± 4.28 ^b^	IW4	82.13 ± 0.97 ^ab^	IW4
3	15.03 ± 19.42 ^c^	IW1	11.13 ± 1.73 ^c^	IW1	79.57 ± 1.97 ^ab^	IW3
4	15.00 ± 11.23 ^c^	AW1	10.89 ± 0.95 ^c^	IW5	75.40 ± 1.32 ^abc^	AW1
5	10.24 ± 3.86 ^cd^	AW3	6.10 ± 0.95 ^cd^	AW1	71.71 ± 2.57 ^bc^	AW5
6	8.70 ± 7.83 ^cd^	AW4	3.10 ± 1.05 ^cde^	AW3	71.55 ± 3.85 ^bc^	AW4
7	3.91 ± 6.58 ^cde^	IW7	0.10 ± 0.94 ^def^	IW2	67.39 ± 7.32 ^c^	IW7
8	3.21 ± 8.27 ^cde^	AW5	−0.44 ± 2.03 ^def^	AW5	66.75 ± 2.18 ^c^	FW2
9	2.08 ± 14.40 ^cdef^	IW3	−2.18 ± 0.75 ^def^	AW2	66.59 ± 2.60 ^c^	AW3
10	2.05 ± 7.41 ^cdef^	AW2	−2.90 ± 2.55 ^def^	IW6	64.34 ± 2.04 ^c^	IW5
11	−0.95 ± 2.41 ^cdefg^	FW2	−3.97 ± 0.92 ^ef^	AW4	51.60 ± 2.47 ^d^	FW1
12	−2.46 ± 8.56 ^defg^	IW5	−5.90 ± 3.83 ^ef^	IW7	47.04 ± 14.65 ^d^	IW6
13	−11.50 ± 5.05 ^efg^	IW2	−8.66 ± 12.68 ^fg^	IW3	46.47 ± 9.94 ^d^	IW2
14	−14.49 ± 1.99 ^fg^	FW1	−15.34 ± 11.78 ^g^	FW2	41.03 ± 11.67 ^d^	AW2
15	−15.03 ± 5.91 ^g^	IW6	−17.56 ± 4.71 ^g^	FW1	24.60 ± 9.02 ^e^	IW1
kanamycin	39.00 ± 2.81		77.60 ± 0.85		87.02 ± 0.42	
ampicillin	31.98 ± 1.59		85.20 ± 10.57		32.77 ± 4.18	

* Inhibitory percentage = [(OD_600_ control − OD_600_ test)/OD_600_ control] × 100. Means ± standard deviations with the same superscript letter in a column were not statistically different (*p* < 0.05) by Duncan’s multiple range test.

**Table 2 antibiotics-12-00448-t002:** Inhibitory percentages of synthetic peptides derived from peptide those purified from protein hydrolysates of bagasse (AW6).

Peptide No.	Accession No.	Protein Name	Peptide Sequence	Inhibitory Percentage		
				*P. aeruginosa*	*Bacillus subtilis*	*Burkholderia cepacia*
**1**	A0A3P3YVC5	Photosystem II CP47 reaction center protein (PSII 47 kDa protein) (Protein CP-47)	GAFHVTGL	21.74 ± 0.96	3.82 ± 0.09	9.09 ± 0.04
**2**	A0A059PYU1	RNA helicase (EC 3.6.4.13)	VLSSWGDESTL	23.61 ± 0.12	4.45 ± 0.03	14.92 ± 0.72
**3**	A0A678TPX2	LRR receptor-like serine/threonine-protein kinase GSO2	NLWSNEINQDMAEF	24.87 ± 0.81	11.33 ± 0.29	23.32 ± 0.68
**4**	A0A059Q010	3-ketoacyl-CoA synthase (EC 2.3.1.)	VSNCL	24.70 ± 0.81	10.39 ± 0.33	20.56 ± 0.28
**5**	A0A059PZS2	Chitinase	WDTDNLSPDAVAAIKAAHPNVAVMAGL	30.34 ± 0.95	5.97 ± 0.06	20.51 ± 0.48

**Table 3 antibiotics-12-00448-t003:** Summary of differentially expressed proteins in human pathogenic bacteria treated with peptide no.3, peptide no.4, and antibiotics.

Human Pathogen	Total Proteins Differentially Expressed	Uniqueness and Commonality of Differentially Expressed Proteins			
		Unique	Shared with Ampicillin	Shared with Kanamycin	Shared with Oxycline
Peptide no.3					
*P. aeruginosa*	198	136	-	-	8
*Bacillus subtilis*	99	39	4	6	9
*Burkholderia cepacia*	4855	436	-	422	375
Peptide no.4					
*P. aeruginosa*	248	186	-	-	7
*Bacillus subtilis*	98	43	4	10	6
*Burkholderia cepacia*	2947	132	-	209	327

**Table 4 antibiotics-12-00448-t004:** Waste sample characteristics; AW, agricultural waste; IW, agro-industrial waste; FW, fishery waste.

Code	Waste Samples	Source of Waste Sample	Location (Latitude, Longitude)
AW1	Rice straw	rice farm	Chachoengsao (13.6690° N, 101.0891° E)
AW2	Corn cobs	corn farm	Sakaeo (13.5035° N, 102.2872° E)
AW3	Corn leaves	corn farm	Sakaeo (13.5035° N, 102.2872° E)
AW4	Corn husks	corn farm	Sakaeo (13.5035° N, 102.2872° E)
AW5	Sugarcane leaves	sugarcane farm	Sakaeo (13.50181° N, 102.2875° E)
AW6	Bagasse	sugarcane farm	Sakaeo (13.50181° N, 102.2875° E)
IW1	Fermented soybeans	light soy sauce productions	Chachoengsao (13.7489° N, 100.9518° E)
IW2	Soybean pellets	soybean milk productions	Chachoengsao (13.6924° N, 101.0807° E)
IW3	Peanut seed coats	peanut-based snack productions	Bangkok (13.6557° N, 100.4305° E)
IW4	Coconut residue	coconut milk productions	Chachoengsao (13.6924° N, 101.0807° E)
IW5	Coffee grounds	Arabica grounds, mainly coffee industry	Chachoengsao (13.6701° N, 101.0562° E)
IW6	Fish residue	fish sauce productions	Bangkok (13.5790° N, 100.4418° E)
IW7	Fish residue (water rinsed)	fish sauce productions	Bangkok (13.5790° N, 100.4418° E)
FW1	Nile tilapia fish fin	fresh market	Chachoengsao (13.6623° N, 101.0343° E)
FW2	*Clarias* sp. Fish fin	fresh market	Chachoengsao (13.6623° N, 101.0343° E)

**Table 5 antibiotics-12-00448-t005:** Guideline for running the 18-well OFFGEL unit with pH gradient 3–10.

Step	Voltage Mode	Voltage (V)	Time (h:min)	kVh
1	Step and hold	500	1:00 (8:00)	0.5
2	Gradient	1000	1:00	0.8
3a	Gradient	8000	3:00	13.5
4a	Step and hold	8000	0:46–1:30	6.2–12.2
3b	Gradient	10,000	3:00	16.5
4b	Step and hold	10,000	0:20–0:55	3.2–9.2
Total				21.0–27.0

## Data Availability

Not applicable.

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
