# Peer review of "Mechanisms of Antimicrobial Peptides from Bagasse against Human Pathogenic Bacteria"

_antibiotics, 2023, doi:10.3390/antibiotics12030448_

Round 1
Reviewer 1 Report
The manuscript “Mechanisms of Antimicrobial Peptides from Bagasse against Human Pathogenic Bacteria” is written well that described the role of antimicrobial peptides to overcome the bacterial growth.
1. The authors should also ensure that each abbreviation is defined at first appearance in this article.
2. Bacterial name should be italics throughout the manuscript.
3. Clearly mentioned the main objective this study in abstract.
4. What was the basis of selection of these species (Pseudomonas aeruginosa, Bacillus subtilis, and Burkholderia cepacian) for antibacterial activity? Why authors did not include drug resistant bacterial isolates. As drug resistant bacteria are major cause of human death. Better to include MDR bacteria and compare the efficacy of AMP with normal pathogens.
5. What’s the distinctive mechanism of AMP. Should be mentioned in abstract.
6. Did authors also check their cytotoxicity against human cells line?
7. what was the MIC and MBC values?
8. Pls provide the agar plates showing zone of inhibition of each tested peptide.
9. Why synthetic peptides were used. Which peptides have better antibacterial synthetic or peptides synthesized from Bagasse?
10. There are some grammar and typo errors. Needs to be checked for spelling and grammatical mistakes like misspelled words, errors with punctuation.
11. The references quoted within the manuscript should be checked for uniformity.
12. Write major findings in conclusion section with future prospects.
Author Response
Response to Reviewer 1 Comments
Point 1: The authors should also ensure that each abbreviation is defined at first appearance in this article.
Response 1: We defined the abbreviation at the first appearance of the word AW, IW, BR, UBR, BC and UBC (Line 84-85, 109-124)
Point 2: Bacterial name should be italics throughout the manuscript.
Response 2: We converted normal text of Scientific name to be italics.
Point 3: Clearly mentioned the main objective this study in abstract.
Response 3: The main objective of this study was already mentioned in Line 75-81 and also in Line 11-13 in abstract.
Point 4: What was the basis of selection of these species (Pseudomonas aeruginosa, Bacillus subtilis, and Burkholderia cepacia) for antibacterial activity? Why authors did not include drug resistant bacterial isolates. As drug resistant bacteria are major cause of human death. Better to include MDR bacteria and compare the efficacy of AMP with normal pathogens.
Response 4: Thank you very much for your suggestion. This is the preliminary identification and characterization of AMPs from agricultural waste. The antimicrobial activity of these AMPs against MDR bacteria and some opportunistic fungi will be studied further.
Point 5: What’s the distinctive mechanism of AMP. Should be mentioned in abstract.
Response 5: As the results from UniProt, these two AMPs have dual mode of action. They act as intracellular-active AMPs and as transmembrane-active AMPs. Then we mentioned this matter in the Line 22-23 in abstract.
Point 6: Did authors also check their cytotoxicity against human cells line?
Point 7: what was the MIC and MBC values?
Response 6-7: The synthetic peptides was not enough to determine their MICs and cytotoxicity against human cells line. The funding and amount of time were also limited. However, MICs or their cytotoxicity against human cells line of these peptides will be determined during modification of these peptides in the future.
Point 8: Pls provide the agar plates showing zone of inhibition of each tested peptide.
Response 8: Broth diltion assay was used to evaluate the antibacterial activity of the synthetic peptides. The inhibition zone of each tested peptide could not be shown.
Point 9: Why synthetic peptides were used. Which peptides have better antibacterial synthetic or peptides synthesized from Bagasse?
Response 9: The peptides from peptic hydrolysate of Bagasse were purified and analyzed for their amino acid sequences by LC-MS/MS. However, many peptides were detected in the active fraction. The peptides expected to have antibacterial activity were then selected to use as representative peptides from Bagasse hydrolysate. The peptide candidates were chemically synthesized and evaluated their antibacterial activity against human pathogens.
Point 10: There are some grammar and typo errors. Needs to be checked for spelling and grammatical mistakes like misspelled words, errors with punctuation.
Response 10: We did reread throughout the manuscript to find typo errors.
Point 11: The references quoted within the manuscript should be checked for uniformity.
Response 11: We have already checked as you suggested.
Point 12: Write major findings in conclusion section with future prospects.
Response 12: We write the major findings and plan in future in the conclusion section. (Line 460-468)

Reviewer 2 Report
In this work, the authors have isolated antimicrobial peptides from agricultural waste and tested them on three different human pathogens namely Pseudomonas aeruginosa, Bacillus sub- 20 tilis, and Burkholderia cepacia.to study their efficacy. Using shot-gun proteomics the authors have dissected out the proteins that got differentially expressed when treated with these peptides.
I’m outlining the comments below:
1. Line 28: Instead of writing species of pathogenic bacteria simply write pathogenic bacteria.
2. Table 1 can be better organized. It looks repetitive. Place all the protein hydrolysates on a single column on the left and their efficacy on three separate columns, one for each of the three pathogenic bacteria tested.
3. The section on the efficacy of the protein hydrolysates after each purification is confusing and needs to be rewritten. What happened to IW4 samples after the cation exchange chromatography? The authors should comment on that.
4. How the authors label the peptides in Table 2 is not clear. The authors should be consistent about naming the peptides. For example, peptide 3 GAFHVTGL is the first peptide in table 1.
5. Figure 2 needs a lot of work. The authors should use the same Venn diagram shapes for all three organisms. For some reason, the authors have used a star shape for Figure 2b for Bacillus subtilis. The authors should replace that to make the figure look consistent. The graph below the Venn diagrams lacks the label for the Y-axis. There is the word “c” in the middle of the figure. The authors should replace it.
6. The authors should provide a list of proteins that got differentially expressed in both treated as well as untreated samples as a supplementary table.
7. Table 3 lacks clarity. For example, the authors claimed that 198 proteins got differentially expressed in Pseudomonas aeruginosa when treated with peptide 3, of which 136 are unique. What are the rest of the proteins? The authors should explain Table 3 clearly in the manuscript.
8. It is surprising that treatment with these antimicrobial peptides did not elicit enough stress response pathways in the three organisms tested. As per Figure 3, only 6.67 % of the proteins were classified as involved with stress response in Bacillus subtilis. The other two organisms didn’t show any upregulation of stress response pathways. One would expect that antimicrobial peptides should generate a huge upregulation of the stress response pathways. as the pathogens are subjected to stress. The authors should comment on that.
9. The authors have not provided any experimental proof of using these antimicrobial peptides on human cell lines. The authors should conduct experiments with human cell lines to show that they do not have any detrimental effects.
Author Response
Response to Reviewer 2 Comments
Point 1: Line 28: Instead of writing species of pathogenic bacteria simply write pathogenic bacteria.
Response 1: We did rewrite as you suggested. (Line 28)
Point 2: Table 1 can be better organized. It looks repetitive. Place all the protein hydrolysates on a single column on the left and their efficacy on three separate columns, one for each of the three pathogenic bacteria tested.
Response 2: Although it looks similar between columns but it’s actually different in order. So we have to place all protein hydrolysates order on every single column as it was.
Point 3: The section on the efficacy of the protein hydrolysates after each purification is confusing and needs to be rewritten. What happened to IW4 samples after the cation exchange chromatography? The authors should comment on that.
Response 3: As shown in figure 1, every sample that came from every single purification step (reverse-phase chromatography, cation exchange chromatography and off gel fractionation) were tested for antibacterial activity, the most effective samples were written in black color while the lower effective sample were written in light grey. The black ones were chosen for further purification step, while the lower ones were not.
For example, after reverse-phase chromatography we got 4 samples; AW6-UBR (unbound fraction from reverse-phase chromatography, AW6-BR (bound fraction), IW4-UBR and IW4-BR. All four samples were tested but only AW6-UBR and IW4-UBR showed the highest antibacterial activity. Then these two samples were chosen for further purification by cation-exchange chromatography.
Point 4: How the authors label the peptides in Table 2 is not clear. The authors should be consistent about naming the peptides. For example, peptide 3 GAFHVTGL is the first peptide in table 1.
Response 4: Thank you so much for your consideration. Peptide no.3 was NLWSNEINQDMAEF and peptide no.4 was VSNCL.
Point 5: Figure 2 needs a lot of work. The authors should use the same Venn diagram shapes for all three organisms. For some reason, the authors have used a star shape for Figure 2b for Bacillus subtilis. The authors should replace that to make the figure look consistent. The graph below the Venn diagrams lacks the label for the Y-axis. There is the word “c” in the middle of the figure. The authors should replace it.
Response 5: About the shape of Venn diagram, we can not change because they were designed by the Venn program software. We thought the shapes depends on number of data list. For example, if data lists less than 6, the result will show in oval shape (like in P. aeruginosa and Burkholderia cepacian) but when data list was 6, the result will show in triangle shape (like in Bacillus subtilis).
About bar graph below Venn diagram, Y-axis is the word “Size of each list”. The excess “C” in the middle of the figure, we removed it.
Point 6: The authors should provide a list of proteins that got differentially expressed in both treated as well as untreated samples as a supplementary table.
Response 6: We provide list of differentially expressed proteins in treated and untreated samples as supplementary tables. The tables are attached with this e-mail.
Point 7: Table 3 lacks clarity. For example, the authors claimed that 198 proteins got differentially expressed in Pseudomonas aeruginosa when treated with peptide 3, of which 136 are unique. What are the rest of the proteins? The authors should explain Table 3 clearly in the manuscript.
Response 7: Table 3 was used to clarify the interesting data from figure 2.
As shown in figure 2a, from 198 proteins, there were only 136 proteins that found only in pathogen treated with peptide no.3. When 8 proteins shared with oxycline, 26 proteins shared with peptide no.4 and 28 proteins shared with control that did not treated with any peptides. In table 3, we selected number that found in only pathogen treated with peptide no3 and antibiotics because we need to focus on materials that might inhibit growth of pathogens (not focus on negative control), and show how much proteins that have similar or different mechanism from antibiotics.
Point 8: It is surprising that treatment with these antimicrobial peptides did not elicit enough stress response pathways in the three organisms tested. As per Figure 3, only 6.67 % of the proteins were classified as involved with stress response in Bacillus subtilis. The other two organisms didn’t show any upregulation of stress response pathways. One would expect that antimicrobial peptides should generate a huge upregulation of the stress response pathways. as the pathogens are subjected to stress. The authors should comment on that.
Response 8: We compared the proteome of peptides treated cells and antibiotics treated cells at 6 h, and found that there was no proteins related to stress pathway in both treatments. This implied that stress response pathway may be triggered by peptides and anitibiotics earlier than 6 h.
Point 9: The authors have not provided any experimental proof of using these antimicrobial peptides on human cell lines. The authors should conduct experiments with human cell lines to show that they do not have any detrimental effects.
Response 9: The synthetic peptides was not enough for checking their cytotoxicity against human cells line. The funding and amount of time were also limit. However, this experiment will be determined during modification of these peptides in the future.

Round 2
Reviewer 1 Report
Authors has revised manuscript satisfactory. I recommend to accept it.
Reviewer 2 Report
Everything looks good